# Replica-Exchange Nosé-Hoover Dynamics
# for Bayesian Learning on Large Datasets

**Rui Luo**[*1], **Qiang Zhang**[*1], **Yaodong Yang**[2,†], **and Jun Wang** [1]
{rui.luo, qiang.zhang, jun.wang}@cs.ucl.ac.uk
{yaodong.yang}@huawei.com
[1]University College London, [2]Huawei R&D U.K.

## Abstract

In this paper, we present a new practical method for Bayesian learning that can rapidly draw representative samples from complex posterior distributions with multiple isolated modes in the presence of mini-batch noise. This is achieved by simulating a collection of replicas in parallel with different temperatures and periodically swapping them. When evolving the replicas' states, the Nosé-Hoover dynamics is applied, which adaptively neutralizes the mini-batch noise. To perform proper exchanges, a new protocol is developed with a noise-aware test of acceptance, by which the detailed balance is reserved in an asymptotic way. While its efficacy on complex multimodal posteriors has been illustrated by testing over synthetic distributions, experiments with deep Bayesian neural networks on large-scale datasets have shown its significant improvements over strong baselines.

## 1 Introduction

Bayesian inference is a principled approach to data analysis and provides a natural way of capturing the uncertainty within the quantities of interest [13]. A practical technique of posterior sampling in Bayesian inference is the Markov chain Monte Carlo methods [15]. Albeit successful in a wide-range of applications, the traditional MCMC methods, such as the Metropolis-Hastings (MH) algorithm [35, 17], the Gibbs sampler [14], and the hybrid/Hamiltonian Monte Carlo (HMC) [9, 38], have significant difficulties in dealing with complex probabilistic models with large datasets. The chief issues are two-fold: first, for large datasets, the exploitation of mini-batches results in noise-corrupted gradient information that drives the sampler to deviate from the correct distributions [7]; second, for complex models, there exists multiple modes, and some might be isolated with others such that the samplers may not discover, leading towards the phenomenon of *pseudo-convergence* [5].

To tackle the first fold of the chief issues, different approaches have been proposed. Several stochastic methods employing the techniques stemmed from molecular dynamics have been devised to alleviate the influence of mini-batch noise, e.g. the stochastic gradient Langevin dynamics (SGLD) [46], the stochastic gradient Hamiltonian Monte Carlo (SGHMC) [7] and the stochastic gradient Nosé-Hoover thermostat (SGNHT) [8]. Alternatively, the classic MH algorithm has been modified to cope with some approximate detailed balance [27, 3, 41]. These methods, however, still suffer from the pseudo-convergence problem.

To address the second fold of the chief issues, the idea of tempering [33, 10, 16] is conceived as a promising framework of solutions. It leverages the finding from statistical physics that a system at high temperature has a better chance to step across energy barriers between isolated modes of the state distribution [29] and hence enables rapid exploration of the state space. Despite the fact that those samplers based on tempering, such as the replica-exchange Monte Carlo [45, 21], the simulated tempering [33] and the tempered transition [37], have shown improvements on sampling complex

---

[*]Equal contribution.

[†]Corresponding author

distributions, the fact that they rely heavily on the exact evaluation of likelihood function, preventing the application on large datasets. Notably, the newly proposed "thermostat-assisted continuously-tempered Hamiltonian Monte Carlo" (TACT-HMC) [31] has attempted to combine the advantage of molecular dynamics and tempering to address the aforementioned chief issues of noise-corrupted gradient and pseudo-convergence. However, its sampling efficiency is relatively low due to the fact that *continuous tempering*, *i.e.* an single-threaded alternative to *replica exchange*, keeps varying the temperature of the inner system; unbiased samples can only be generated when the inner system stays at the unity temperature, which corresponds to only a fraction of entire simulation interval.

To address altogether every facets of the chief issues, *i.e.* the mini-batch gradient and pseudo-convergence, with a higher tempering efficiency, we propose a new sampler, *Replica-exchange Nosé-Hoover Dynamics* (RENHD). Our method simulates in parallel a collection of replicas with each at a different temperature. It automatically neutralizes the noise arising from mini-batch gradients, by equipping the Nosé-Hoover dynamics [11] for each replica. To alleviate pseudo-convergence, RENHD periodically swaps the configurations of replicas, during which a noise-aware test of acceptance is used to keep the detailed balance in an asymptotic fashion. As for tempering efficiency, our approach keep monitoring the replicas at unity temperature whilst exploring the parameter space in high temperatures, which constantly generate unbiased samples.

Compared to the existing methods, the novelty of RENHD lies in 1) it is the **first** *"practical"* replica-exchange solution applicable to mini-batch settings for large datasets 2) the integration of Nosé-Hoover dynamics with replica-exchange framework to enable rapid generation of unbiased samples from complex multimodal distributions, especially when deep neural nets are applied; 3) the elaboration of the noise-aware exchange protocol with Gaussian deconvolution, providing an analytic solution that improves the exchange efficiency and reproducibility. By the term *"practical"*, we refer to the merit of our solution that it can be readily implemented, deployed, and utilized. The integration leads to an ensemble of tempered replicas, each resembling an instance of "tempered" SGNHT. Experiments are conducted to validate the efficacy and demonstrate the state-of-the-art performance in Bayesian posterior sampling tasks with complex multimodal distributions and large datasets; it outperforms the classic baselines by a significant improvement on the accuracy of image classification. Notably, the experiment shows that RENHD enjoys a much higher efficiency in generating unbiased samples. Within same real-world time of execution, RENHD produces 4 times as many unbiased multimodal samples as by the tempered alternative TACT-HMC.

## 2 Replica-exchange Nosé-Hoover Dynamics

This section gives a detailed description of our *Replica-exchange Nosé-Hoover Dynamics* (RENHD). This method constains two alternating subroutines: 1.) dynamics evolution of all replicas in parallel, and 2.) configuration exchange between adjacent replicas when detailed balance is met.

### 2.1 Evolving replicas using the Nosé-Hoover dynamics with mini-batch gradient

A standard recipe for generating representative samples from $\rho(\theta|\mathscr{D})$ begins with establishing a mechanical system with a point mass moving in $d$-dimensional Euclidean space. The variable of interest $\theta$ is called the system configuration, indicating the particle's position. The target posterior transforms into the potential field $U(\theta) := -\log\rho(\theta|\mathscr{D}) + \text{const}$, which defines the energy landscape of the physical system. Intuitively, the force induced by $U(\theta)$ guides the motion of the particle, tracing out the trajectory $\theta(t)$; the snapshots $\{\theta_k\}$ registered periodically from $\theta(t)$ will be examined and accepted probabilistically as new samples.

As the entire dataset $\mathscr{D}$ is involved in calculating the force $f := -\nabla U$, it becomes computationally very expensive or even infeasible when $\mathscr{D}$ grows large. Therefore, for practical purpose, we resort to mini-batches $\mathcal{S} \subset \mathscr{D}$ for big datasets, resulting in noisy estimates approximating the actual force

$$\tilde{f}(\theta) := \nabla\log\rho(\theta) + \frac{|\mathscr{D}|}{|\mathcal{S}|}\sum_{x\in\mathcal{S}}\nabla\log\ell(\theta;x) \approx f(\theta). \tag{1}$$

It is clear that $\tilde{f}$ is an unbiased estimator of $f$ given that each $x \in \mathscr{D}$ is *i.i.d.* As sum of independent random variables, $\tilde{f}$ converges to Gaussian asymptotically by Central Limit Theorem (CLT) such that

$$\tilde{f}(\theta)\,dt = f(\theta)\,dt + \mathcal{N}(0, 2B\,dt) \quad \text{with } B := \mathbf{var}[\tilde{f}]\,dt/2 \text{ defining the noise intensity.} \tag{2}$$

We assume the variance of $\tilde{f}$ being a constant value due to its $\theta$-independence verified by [6] and isotropic in all dimensions for $\theta$'s symmetric nature as suggested in Ding et al.'s scheme [8].

Now we construct an increasing ladder of temperatures $\{T_j\}$. On each rung $j$, we instantiate a replica of the physical system established previously. For each replica $j$, a set of dynamic variables is defined, which we refer to as the system state $\Gamma_j = (\theta_j, p_j, \xi_j)$, with $p_j \in \mathbb{R}^d$ being $\theta_j$'s conjugate momentum and $\xi \in \mathbb{R}$ denoting the Nosé-Hoover thermostat [39, 20] for adaptive noise dissipation [24]. After configuring unity mass, there is only *one* "replica-specific" constant to be further assigned, *i.e.* the temperature $T_j$. We define the time evolution of $\Gamma_j$ as a variant of the Nosé-Hoover dynamics [11]:

$$\mathrm{d}\theta_j = p_j\,\mathrm{d}t, \quad \mathrm{d}p_j = \left[\tilde{f}(\theta_j) - (\xi_j + \chi)p_j\right]\mathrm{d}t + \mathcal{N}(0, 2\chi T_j\,\mathrm{d}t), \quad \mathrm{d}\xi_j = \left[p_j^\top p_j - T_j d\right]\mathrm{d}t. \quad (3)$$

where $\chi \in \mathbb{R}_+$ is the intensity determining the Langevin background noise. This variant in (3) is originally proposed as the *"adaptive Langevin dynamics"* (Ad-Langevin) [24], in which the vanilla Nosé-Hoover dynamics is blended with the second-order Langevin dynamics. Essentially, we have both the time-varying Nosé-Hoover thermostat $\xi_j$ for adaptive adjustment for the dynamics in the presence of stochastic gradient noise, and the time-invariant Langevin background noise that promotes replicas' ergodicity in simulation. The following theorem validates the efficacy of dynamics in (3).

**Theorem 1** ([24]). *The dynamics defined in* (3) *ensures the distribution of* $\Gamma_j$ *converging to the unique stationary distribution*

$$\pi_j(\Gamma_j) \propto e^{-\frac{U(\theta_j)+p_j^\top p_j/2}{T_j}} \exp\left[-\frac{(\xi_j - B/T_j)^2}{2T_j}\right], \quad (4)$$

*if $j$ is ergodic. The intensity B of gradient noise is in* (2).

*Proof.* The details can be found in Appendix. □

We simulate the time evolution of all replicas $\{j\}$ in parallel using the dynamics in (3) until converged. A quick observation on (4) reveals the fact that all replicas share the same functional form of $\pi_j(\Gamma_j)$; the discrepancy between one invariant distribution and another is merely the result of different temperatures. Considering replica $j$ at temperature $T_j$, one can easily obtain the invariant distribution of $\theta_j$ by marginalizing (4) *w.r.t.* $p_j$ and $\xi_j$:

$$\pi_j(\theta_j) \propto \int \pi_j(\Gamma_j)\,\mathrm{d}p_j\,\mathrm{d}\xi_j \propto e^{-U(\theta_j)/T_j}, \quad (5)$$

where the "effective" potential at $T_j$ is essentially the actual potential $U$ rescaled by a factor of $1/T_j$. A physical interpretation is that the energy barriers separating isolated modes are effectively lowered and hence easier to overcome at high temperatures. Consequently, replicas at higher $T$'s enjoy more efficient exploration of $\theta$-space. On the other hand, for the "replica 0", *i.e.* the one at unity temperature $T_0 = 1$, the marginal reads $\pi(\theta) \propto e^{-U(\theta)} \propto \rho(\theta|\mathcal{D})$, recovering the target posterior.

**Comparison to Langevin dynamics.** As an alternative to the Nosé-Hoover dynamics, the Langevin dynamics, or the Langevin equation [30] in physics, lays the foundation of Langevin's approach to modeling molecular systems with stochastic perturbations. Langevin dynamics in its complete form is presented as a second-order stochastic differential equation, which is equivalent to the dynamics described in SGHMC. Its complexity in simulation is roughly the same as the Nosé-Hoover's equation system in (3). SGLD, on the other hand, uses the overdamped Langevin equation, *i.e.* a simplified version with mass limiting towards zero. Albeit simpler, SGLD can be drastically slower in exploring distant regions in $\theta$-space due to its random-walk-like updates [7], rendering it inapt for multimodal sampling. The major advantage of the Nosé-Hoover dynamics over Langevin's approach, is that the former method adaptively neutralizes the effect of noisy gradient using a simple dynamic variable whereas the latter requires an additional subroutine for noise estimation, which is expensive and needs to be performed manually [2]. In other words, the Nosé-Hoover thermostat saves us from expensive manual noise estimation with the least cost; it works well with minimal prior knowledge.

## 2.2 Exchanging replicas by logistic test with mini-batch estimates of potential differences

As we have investigated, replicas at high temperatures have better chances to transit between modes, leading to faster exploration of $\theta$-space. However, such advantage comes at a price that the sampling is no longer correct: the spectrum of sampled distribution widens in proportional to the square root of replica's temperature. It implies that the samples shall only be drawn at unity temperature.

To enable rapid $\theta$-space exploration for high-temperature replicas while retaining accurate sampling, a new protocol is devised that periodically swaps the configurations of replicas, with the compatibility of mini-batch settings; the term "replica exchange" refers to the operations that swap $\theta$'s.

The protocol, as a non-physical probabilistic procedure, is built on the principle of detailed balance, *i.e.* every exchange shall be equilibrated by its reverse counterpart, or formally

$$\pi_j(\theta_j)\pi_k(\theta_k)\alpha_{jk}\big[(\theta_j,\theta_k) \to (\theta_k,\theta_j)\big] = \pi_j(\theta_k)\pi_k(\theta_j)\alpha_{jk}\big[(\theta_j,\theta_k) \leftarrow (\theta_k,\theta_j)\big]. \qquad (6)$$

The left side corresponds to a forward exchange and the right side is for its backward counterpart. $\pi_j$ in (6) represents the probability of replica $j$ being in some certain configuration (*cf.* (5)), and $\alpha_{jk}$ is the probability of a successful exchange (either forward or backward) between the configurations of the pair of two replicas $(j,k)$. Such probability is determined by a test of acceptance, which is a criterion associated with the ratio of probabilities $\pi_j(\theta_k)\pi_k(\theta_j)\big/\pi_j(\theta_j)\pi_k(\theta_k)$. Given (5), the probability ratio can be calculated from the potential difference $\Delta E_{jk} := \big[U(\theta_j) - U(\theta_k)\big]\big[1/T_j - 1/T_k\big]$.

When switching to mini-batches, the actual potential difference $\Delta E_{jk}$ is no longer accessible; instead, we only obtain a noisy estimate of the potential difference

$$\Delta \tilde{E}_{jk} := \left[\frac{1}{T_j} - \frac{1}{T_k}\right]\left[\log\frac{\rho(\theta_j)}{\rho(\theta_k)} + \frac{|\mathcal{D}|}{|\mathcal{S}|}\sum_{x\in\mathcal{S}}\log\frac{\ell(\theta_j;x)}{\ell(\theta_k;x)}\right], \qquad (7)$$

which is essentially a random variable centered at $\Delta E_{jk}$. Moreover, $\Delta \tilde{E}_{jk}$ converges asymptotically to Gaussian as indicated by CLT; the following factorization is applicable

$$\Delta \tilde{E}_{jk} = \Delta E_{jk} + z_{\mathcal{N}}, \ z_{\mathcal{N}} \sim \mathcal{N}(0,\sigma^2), \ \text{and } \sigma^2 := \mathbf{var}[\Delta \tilde{E}_{jk}]. \qquad (8)$$

Now we introduce an auxiliary variable $z_{\mathscr{C}}$ to fix the corrupted estimate $\Delta \tilde{E}_{jk}$; its distribution, which we refer to as the *compensation distribution*, is formulated as

$$\tilde{q}_{\mathscr{C}}(z) \propto \sum_{n=0}^{\infty}\frac{(-1)^n}{\lambda^n n!}\cdot H_n\left[\frac{\lambda\sigma^2}{4}\right]\cdot g^{(2n+1)}(z), \ \text{ with } \sigma^2 = \mathbf{var}[\Delta \tilde{E}_{jk}], \qquad (9)$$

where $H_n[\cdot]$ represents the Hermite polynomials [1] and $g := 1/1+e^{-z}$ defines the logistic function. We denote $g^{(k)} := \mathrm{d}^k g/\mathrm{d}z^k$ as the $k$-th derivative of $g$. The argument $\lambda$ controlling the "bandwidth" is crucial according to the following theorem.

**Theorem 2.** *Given a pair of replicas $(j,k)$ currently being at the configurations $(\theta_j,\theta_k)$, the exchange $(\theta_j,\theta_k) \to (\theta_k,\theta_j)$ preserves formally the condition of detailed balance, if one admits the attempt of exchange under the criterion*

$$z_{\mathscr{C}} + \Delta \tilde{E}_{jk} > 0, \qquad (10)$$

*where $z_{\mathscr{C}} \sim \tilde{q}_{\mathscr{C}}$ in (9) and evaluated in the limit of $\lambda \to +\infty$. The estimate $\Delta \tilde{E}_{jk}$ is defined in (7).*

*Proof.* Given Barker's acceptance test [4], the acceptance probability for detailed balance (6) reads

$$\alpha_{jk}^{\mathrm{B}}\big[(\theta_j,\theta_k) \to (\theta_k,\theta_j)\big] := 1\big/\big[1+e^{-\Delta E_{jk}}\big].$$

Note that $\alpha_{jk}^{\mathrm{B}}$ is in the form of the logistic function $g$ in $\Delta E_{jk}$, which leads to the standard logistic distribution $\mathscr{L}(0,1)$. The corresponding criterion working with full-batch potential difference $\Delta E_{jk}$

$$z_{\mathscr{L}} + \Delta E_{jk} > 0, \ \text{with } z_{\mathscr{L}} \sim \mathscr{L}(0,1). \qquad (11)$$

When using mini-batches, the noisy estimate $\Delta \tilde{E}_{jk}$ kicks in as "corrupted" measurements of the actual value $\Delta E_{jk}$ under Gaussian perturbations $z_{\mathcal{N}}$ as derived in (8). The mini-batch criterion is devised by decomposing the logistic variable $z_{\mathscr{L}}$ in (11) as suggested by Seita *et al.* [41]:

$$z_{\mathscr{L}} + \Delta E_{jk} > 0 \implies z_{\mathscr{C}} + \Delta \tilde{E}_{jk} > 0 \quad \text{with } \Pr[z_{\mathscr{L}} > -\Delta E_{jk}] = \Pr[z_{\mathscr{C}} + z_{\mathcal{N}} > -\Delta E_{jk}], \qquad (10')$$

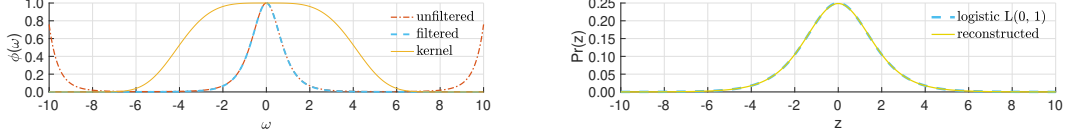

Figure 1: (*colored*) The *left* subplot shows the divergence (*red*) of real spectrum ratio in evaluating (13); it begins to diverge at $|\omega| > 7$. A super-smooth kernel (*solid orange*) is applied by multiplication on the ratio and the composite spectrum (*blue*) is guaranteed to converge. The *right* one compares the desired standard logistic density (the left side of (12), *dashed blue*) with the reconstruction (the right side of (12), *solid yellow*), indicating a good precision. The variance of the Gaussian perturbation is $\sigma^2 = 0.2$ while the bandwidth is $\lambda = 10$; the reconstructed density with first 3 terms in the series (9).

where the new variable $z_{\mathscr{C}}$ *compensates* the discrepancy between the logistic variable for the test $z_{\mathscr{L}}$ and the Gaussian perturbations $z_{\mathscr{N}}$, and the factorization in (8) is applied.

The compensation distribution $q_{\mathscr{C}}$ for $z_{\mathscr{C}}$ is defined in an implicit way by the convolution equation

$$q_{\mathscr{L}}(z) = q_{\mathscr{C}}(z) * q_{\mathscr{N}_{\sigma^2}}(z), \tag{12}$$

where $q_{\mathscr{L}}$ defines the standard logistic density $\mathscr{L}(0, 1)$ while $q_{\mathscr{N}_{\sigma^2}}$ is the Gaussian $\mathscr{N}(0, \sigma^2)$.

The convolutional relation (12) defines a Gaussian deconvolution problem [23] *w.r.t.* the standard logistic distribution, where the Fourier transform can be applied, converting the convolution of densities into the corresponding algebraic equation of characteristics functions:

$$\phi_{\mathscr{L}}(\omega) = \phi_{\mathscr{C}}(\omega) \cdot \phi_{\mathscr{N}_{\sigma^2}}(\omega) \quad \text{where} \quad \phi_{\mathscr{L}}(\omega) = \frac{\pi\omega}{\sinh \pi\omega} \quad \text{and} \quad \phi_{\mathscr{N}_{\sigma^2}}(\omega) = \exp\left[-\frac{\sigma^2\omega^2}{2}\right]. \tag{13}$$

Due to the fact that the Gaussian characteristic function $\phi_{\mathscr{N}_{\sigma^2}}$ decays much faster than the logistic counterpart $q_{\mathscr{L}}$ at high frequencies, the spectrum of $\phi_{\mathscr{C}}$ diverges at infinity such that direct approaches might not apply. Therefore, we design a symmetric multiplicative kernel $0 < \psi_\lambda(\omega) \leq 1$ peaked at the origin with rapid decay; it is parameterized by the "bandwidth" $\lambda > 0$. The inverse Fourier transform with kernels is shown as (see [12])

$$\tilde{q}_{\mathscr{C}}(z) := \mathscr{F}^{-1}\left[\psi_\lambda(\omega) \cdot \phi_{\mathscr{C}}(\omega)\right] = \frac{1}{2\pi} \int_{-\infty}^{\infty} \left[\frac{\psi_\lambda(\omega)}{\phi_{\mathscr{N}_{\sigma^2}}(\omega)}\right] \phi_{\mathscr{L}}(\omega) e^{-iz\omega} \, \mathrm{d}\omega, \tag{14}$$

where $\psi_\lambda(\omega) := e^{-\omega^4/\lambda^2}$ serves as a "low-pass filter" selecting components within a finite bandwidth and suppressing high frequencies so that the spectrum of $\phi_{\mathscr{C}}$ will eventually vanish at infinity and an solution can therefore be guaranteed.

To solve for an series solution to $\tilde{q}_{\mathscr{C}}$, consider the ratio in the brackets in (14), it is the generating function of Hermite polynomials $H_n[u]$ (see [1]):

$$\frac{\psi_\lambda(\omega)}{\phi_{\mathscr{N}_{\sigma^2}}(\omega)} = \frac{e^{-\omega^4/\lambda^2}}{e^{-\sigma^2\omega^2/2}} = \exp\left[2\left(\frac{\lambda\sigma^2}{4}\right) \cdot \left(\frac{\omega^2}{\lambda}\right) - \left(\frac{\omega^2}{\lambda}\right)^2\right] = \sum_{n=0}^{\infty} \frac{1}{\lambda^n n!} \cdot H_n\left[\frac{\lambda\sigma^2}{4}\right] \cdot \omega^{2n}, \tag{15}$$

where we utilized the exponential generating function with $H_n[u]$: $\exp(2ut - t^2) = \sum_{n=0}^{\infty} H_n[u] \, t^n/n!$. Now we substitute (15) back into (14) and rearrange terms, which results in the series solution:

$$\tilde{q}_{\mathscr{C}}(z) \propto \frac{1}{2\pi} \int_{-\infty}^{\infty} \left[\frac{1}{\lambda^n n!} \cdot H_n\left[\frac{\lambda\sigma^2}{4}\right] \cdot \omega^{2n}\right] \phi_{\mathscr{L}}(\omega) e^{-iz\omega} \, \mathrm{d}\omega$$

$$= \sum_{n=0}^{\infty} \left\{\frac{(-1)^n}{\lambda^n n!} \cdot H_n\left[\frac{\lambda\sigma^2}{4}\right] \cdot \left[\frac{1}{2\pi} \int_{-\infty}^{\infty} (i\omega)^{2n} \phi_{\mathscr{L}}(\omega) e^{-iz\omega} \, \mathrm{d}\omega\right]\right\}$$

$$= \sum_{n=0}^{\infty} \frac{(-1)^n}{\lambda^n n!} \cdot H_n\left[\frac{\lambda\sigma^2}{4}\right] \cdot \frac{\mathrm{d}^{2n}}{\mathrm{d}z^{2n}} q_{\mathscr{L}}(z) = \sum_{n=0}^{\infty} \frac{(-1)^n}{\lambda^n n!} \cdot H_n\left[\frac{\lambda\sigma^2}{4}\right] \cdot g^{(2n+1)}(z), \tag{16}$$

where in equity (16), we exploited the nice property of the Fourier transform regarding derivatives. Note that $g(z)$ is the logistic function defined in (9).

As $\lambda$ grows larger, the kernel gains a higher bandwidth, and the series solution $\tilde{q}_{\mathscr{C}}$ gets closer in an asymptotic manner to the improper compensation distribution $q_{\mathscr{C}}$ defined in (12). In other words, in the limit of infinite bandwidth $\lambda \to +\infty$, the formal compensation distribution is achieved, *i.e.* $\tilde{q}_{\mathscr{C}} \to q_{\mathscr{C}}$; the detailed balance is ensured under the criterion (10) with $z_{\mathscr{C}} \sim \tilde{q}_{\mathscr{C}}$ in (9). □

---

**Algorithm 1** The replica-exchange protocol

---

1: **function** EXCHANGE($\theta_j, \theta_k, \texttt{model}, \mathcal{D}, |\mathcal{S}|_{\text{re}}, \sigma_*^2, \hat{q}_{\mathscr{C}}$)
2:     **repeat**
3:         $\mathcal{S} \leftarrow \left[\mathcal{S}, \text{NEXTBATCH}(\mathcal{D}, |\mathcal{S}|_{\text{ex}})\right]$ then eval $\Delta \tilde{E}_{jk}$ by (7) with $\mathcal{S}$, $\texttt{model}.\rho()$, $\texttt{model}.\ell()$
4:     **until var**$[\Delta \tilde{E}_{jk}] < \sigma_*^2$
5:     $z_{\mathcal{N}_*} \sim \mathcal{N}(0, \sigma_*^2 - \textbf{var}[\Delta \tilde{E}_{jk}])$                    ▷ ensuring total **var** $\approx \sigma_*^2$
6:     $z_{\mathscr{C}} \sim \hat{q}_{\mathscr{C}}$                                                 ▷ Gibbs or HMC on $\hat{q}_{\mathscr{C}}$
7:     **if** $z_{\mathscr{C}} + z_{\mathcal{N}_*} + \Delta \tilde{E} > 0$ **then**:                 ▷ checking criterion (17)
8:         $(\theta_j, \theta_k) \leftarrow (\theta_k, \theta_j)$                 ▷ swapping configurations

---

Instead of launching brutal-force attack with an arsenal of numerical solvers, we have provided with an analytic treatment that is more efficient and easier to reproduce. In practice, for any level of precision, we can always find a suitable finite bandwidth $\lambda < +\infty$, with which the analytic series solution $\tilde{q}_{\mathscr{C}}$ in (9) maintains a desired precision of approximation to the actual improper $q_{\mathscr{C}}$. Figure 1 provides with a illustration of the effect of the multiplicative kernel and a comparison between the desired standard logistic density and the reconstruction of convolving the Gaussian perturbations with the series solution $\tilde{q}_{\mathscr{C}}$ calculated in (9).

**Preference of logistic test.** The advantage of Barker's logistic test over its Metropolis counterpart lies in the super-smooth nature of the logistic function. Smoothness ensures the existence of smooth derivatives of infinite orders, which facilitates analytic formulations with infinite series, especially for problems involving deconvolutions. The Metropolis test, albeit more efficient [17], is composed by non-smooth operations, which inevitably introduces Delta functions that sabotage the analyticity.

**Selection of the bandwidth $\lambda$.** The bandwidth $\lambda$ governs the accuracy of sampling by determining the quality of the approximation $\tilde{q}_{\mathscr{C}}(z)$ on the compensation distribution: with higher $\lambda$, $\tilde{q}_{\mathscr{C}}(z)$ becomes more precise; with better $\tilde{q}_{\mathscr{C}}(z)$, the detailed balance can be better preserved, which will lead to more accurate samples. Indeed, higher $\lambda$ results in more time-consuming computation, but we noticed that with relatively small sample variance $\sigma^2$, as is often the case in practice, the accuracy and the computational complexity can be well-balanced by assigning $\lambda$ a moderate value.

### 2.3 Revisiting the assumption of Gaussianity

The assumption of Gaussianity lays the foundation of Nosé-Hoover dynamics; it is often assumed *a priori* in the name of CLT [32]. Recently, some critiques arise about this assumption: the Gaussianity of mini-batch gradients in training AlexNet [28] seems to be complicated as revealed by the tail index analysis [43]. It turns out that for AlexNet and the fully-connected networks, the gradient noise shows some phenomena of heavy-tailed random variables. On the other hand, we examined some other architectures, namely the residual networks (ResNet) [18] as well as the classic LSTM architecture [19], using a conventional recipe, *i.e.* Shapiro-Wilk algorithm [42]. The result indicates for those architectures, the Gaussianity of gradient noise is well-preserved with typical batch sizes in practice (see Appendix C). The following experiments are conducted using those architectures with Gaussianity preserved, because the proposed method is built on Gaussian noise.

## 3 Implementation

This section is devoted to the implementation of RENHD in practical scenarios. First, we devise the replica-exchange protocol based on Theorem 2. Particular attention will be paid to the computation of the series solution $\tilde{q}_{\mathscr{C}}$ in (9). Recall $\tilde{q}_{\mathscr{C}}$, it depends on two parameters: the variance of mini-batch estimate $\sigma^2 = \textbf{var}[\Delta \tilde{E}_{jk}]$ and the bandwidth of kernel $\lambda$. For each attempt of exchange, the mini-batch is different, thus $\tilde{q}_{\mathscr{C}}$ needs re-evaluation due to the flunctuations of the variance $\sigma^2$.

To avoid the consuming re-computation of $\tilde{q}_{\mathscr{C}}$, we would like to reuse a fixed $\hat{q}_{\mathscr{C}}$ throughout the entire sampling process. We set a mild variance threshold $\sigma_*^2$ and an appropriate bandwidth $\lambda$, then compute $\hat{q}_{\mathscr{C}}$ before sampling. During the process, when an attempt of exchange is proposed, we make sure the variance of estimates is below the threshold $\textbf{var}[\Delta \tilde{E}_{jk}] < \sigma_*^2$ by enlarging the mini-batches, and then make up for the difference $\sigma_*^2 - \textbf{var}[\Delta \tilde{E}_{jk}]$ using an additive Gaussian noise such that the

---
**Algorithm 2** Replica-exchange Nosé-Hoover dynamics
---
1: **function** NHDYNAMICS($\{\theta_j\}, \{T_j\}, \texttt{model}, \mathscr{D}, |\mathcal{S}|_{\text{nhd}}, N, \epsilon, c$)  ▷ NHD length $N$; $\epsilon, c$ in (20)
2:  **for all** $\{j\}$ **do**  ▷ all $j$ running in parallel
3:   $v_j \sim \mathcal{N}(0, T_j\epsilon)$ and $s_j \leftarrow c/T_j$
4:   **for** $n = \text{RANGE}(1, N)$ **do**
5:    $\mathcal{S} \leftarrow \text{NEXTBATCH}(\mathscr{D}, |\mathcal{S}|_{\text{nhd}})$
6:    $\tilde{f}_j \leftarrow \texttt{model.BACKWARD}(\theta_j, \mathcal{S})$  ▷ see (1)
7:    $v_j \leftarrow v_j + \tilde{f}_j\epsilon - s_jv_j + \mathcal{N}(0, 2c\epsilon)$  ▷ see (19)
8:    $\theta_j \leftarrow \theta_j + v_j$ then $s_j \leftarrow s_j + \left[v_j^\top v_j/d - T_j\epsilon\right]$
9:  **return** $\{\theta_j\}$

10: MAIN:
11: $\{\theta_j\} \leftarrow \text{RANDN}()$  ▷ initialization
12: $\texttt{args} \leftarrow \left(|\mathcal{S}|_{\text{nhd}}, N, \epsilon, c\right)$  ▷ packing arguments
13: **loop**
14:  $\{\theta_j\} \leftarrow \text{NHDYNAMICS}(\{\theta_j\}, \{T_j\}, \texttt{model}, \mathscr{D}, \texttt{args})$
15:  $\{(j, k)\} \leftarrow \text{RAND}()$  ▷ replicas to swap
16:  **for all** $\{(j, k)\}$ **do**
17:   $\text{EXCHANGE}(\theta_j, \theta_k, \texttt{model}, \mathscr{D}, |\mathcal{S}|_{\text{re}}, \sigma_*^2, \tilde{q}_{\mathscr{C}})$
18:  $\texttt{samples} \leftarrow \left[\texttt{samples}, \theta_0\right]$  ▷ $\theta_0$ from *replica 0* for true posterior
---

overall variance is exactly $\sigma_*^2$. The modified criterion is hence

$$z_{\mathscr{C}} + z_{\mathcal{N}_*} + \Delta\tilde{E} > 0, \quad \text{with } z_{\mathscr{C}} \sim \hat{q}_{\mathscr{C}}, \ z_{\mathcal{N}_*} \sim \mathcal{N}(0, \sigma_*^2 - \mathbf{var}[\Delta\tilde{E}_{jk}]). \tag{17}$$

Intuitively, with $\Delta\tilde{E}_{jk}$ evaluated in (7), one draws $z_{\mathscr{C}}$ from $\hat{q}_{\mathscr{C}}$ and the Gaussian noise $z_{\mathcal{N}_*}$ from $\mathcal{N}(0, \sigma_*^2 - \mathbf{var}[\Delta\tilde{E}_{jk}])$, then examines the sign of $z_{\mathscr{C}} + z_{\mathcal{N}_*} + \Delta\tilde{E}$; the attempt of exchange is accepted when the sum gives a positive outcome. Algorithm 1 summarizes the protocol.

Given $\sigma_*^2$ and $\lambda$, we can pin down each of the terms within the series (9). There is a nice property of the logistic derivatives $g^{(k)}$ that all $g$'s derivatives can be formulated as polynomials in terms of $g$ itself, and coefficients is extracted from the *Worpitzky Triangle*. In Appendix B, we compute the first three Hermite polynomials as well as the logistic derivatives of odd orders.

The parameter setting of experiment is $\sigma_*^2 = 0.2$ and $\lambda = 10$. We truncate the infinite series in (9) and takes the first 3 terms to assemble an approximated solution. The compensation is implemented as

$$\hat{q}_{\mathscr{C}} \approx 0.895g - 0.145g^2 - 2.1g^3 + 2.55g^4 - 1.8g^5 + 0.6g^6, \tag{18}$$

where $g(z) = 1/[1 + e^{-z}]$ denotes the logistic function. Note that we have conducted numerical evaluation on truncated series: with appropriate $\sigma_*^2$ and $\lambda$, the convergence of (18) with 3 terms is fast; using more terms is feasible but may not be advantageous by overall computation cost.

Compared with the numerical treatment proposed by [41], our analytic approach is easier to reproduce and much faster to sample: for any given precision, one can readily re-compute the compensation distribution by using more terms of Hermite polynomials and logistic derivatives (see Appendix B), instead of invoking the entire numerical procedure. Empirical evaluation shows 20× acceleration in sampling using our analytic approach with the Gibbs sampler; the numerical solution in comparison uses the pre-computed density[3] and the conventional methods, *i.e.* binary search and hash tables.

Now we turn to the implementation of the Nosé-Hoover dynamics and the temperature ladder for replicas to run. Recall the dynamics in (3), the inherit noise of mini-batch gradient can be seperated by the Nosé-Hoover thermostat. And the correct canonical distribution can be recovered as stated in Theorem 1, if the system is *ergodic*. However, there are some concerns regarding the ergodicity of the Nosé-Hoover dynamics [34]. We alleviate this issue by introduce additive Gaussian noise such that the dynamics becomes more "stochastic". So we modify the dynamics for momentum $p$ in (3) as

$$\mathrm{d}p_j = \left[\tilde{f}(\theta_j) - \xi_jp_j\right]\mathrm{d}t + \mathcal{N}(0, 2C\,\mathrm{d}t), \tag{19}$$

where the additive Gaussian noise has a pre-defined, constant intensity $C > 10B$ in (2). With non-vanishing time steps $\Delta t$, we make a change of variables for each replica $j$:

$$\text{variables} \quad v_j := p_j\Delta t, \ s_j := \xi_j\Delta t, \qquad \text{constants} \quad \epsilon := \Delta t^2, \ c := C\Delta t. \tag{20}$$

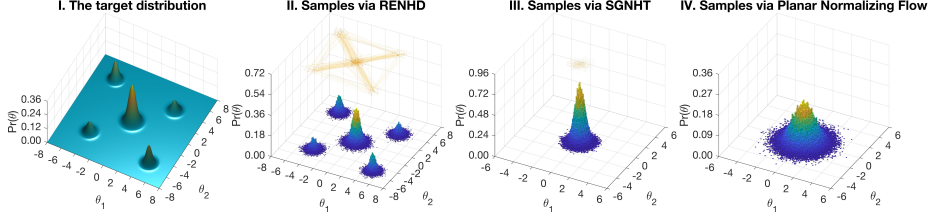

Figure 2: Experiment on sampling a 2$d$ mixture of 5 Gaussians.

For the temperature ladder, we prefer a simpler scheme with the surest bet that the temperature on each rung increases geometrically as suggested by [26] and [36] so that the ladder $\{T_j\}$ of $M$ rungs is

$$T_j = \tau^j \text{ with } \tau > 1 \text{ and } j = 1, 2, \ldots, M. \tag{21}$$

Algorithm 2 describes the implementation of RENHD.

# 4 Experiment

We conduct two sets of experiments: the first uses synthetic distributions, validating the properties of RENHD; the second is on real datasets, showing drastic improvement in classification accuracy.

## 4.1 Synthetic distributions

To validate the efficacy of RENHD, we perform a sampling test on a synthetic 2$d$ Gaussian mixture with 5 isolated modes. The potential energy and its gradient is perturbed by zero-mean Gaussian noise with variance $\sigma^2 = 0.25$ which stays unknown for samplers. A temperatures ladder is established with $M = 7$ rungs and geometric factor $\tau = 1.5$, We compare the sampled histogram with SGNHT, as a non-tempered alternative, and Normalizing Flow (NF) [40], which is a typical variational method. Figure 2 demonstrates that REHND has accurately sampled the target multimodal distribution in the presence of mini-batch noise. On the contrary, SGNHT and NF failed to discover the isolated modes; the latter deviates severely due to the noise, resulting in a spread histogram. We have depicted the sampling trajectory above for the samplers, indicating a good mixing property of RENHD against SGNHT. The effective sample size (ESS) of RENHD is $4.1638 \times 10^3 / 10^5$.

## 4.2 Bayesian learning on real images with convolutional and recurrent neural networks

We run two tasks of image classification on real datasets: Fashion-MNIST on a recurrent neural network and CIFAR-10 on a residual network (ResNet) [18], where we are focusing on finding good modes on multimodal posteriors. The performance is compared on the accuracy of classification.

**Baselines.** SGLD, SGHMC, SGNHT, and TACT-HMC are chosen as alternatives of Bayesian samplers; the SGD with momentum [44] as well as Adam [25] are compared, as the typical methods for training neural nets. To validate the efficacy of the Nosé-Hoover dynamics, we devise the "replica-exchange Langevin dynamics" (RELD) with the same tempering setting by freezing the thermostat $s_j$ in RENHD at the value of $0.999 + c/T_j$, approximating a tempered version of SGLD for a very light particle. All 7 baselines are tuned to their best on each task; the samplers' accuracy is calculated from Monte Carlo integration on sampled models; the optimizers are evaluated directly after training.

**Settings.** For all methods, a single run has 1000 epochs. Random permutation is applied to a percentage (0%, 20%, or 30%) of the training labels, at the beginning of each epoch as suggested by [31] to increase the model uncertainty. We set the mini-batch size $|\mathcal{S}|_{\mathrm{nhd}} = 128$ for the Nosé-Hoover dynamics and $|\mathcal{S}|_{\mathrm{re}} = 256$ for the exchange protocol. The ladder is built with $M = 12$ rungs with geometric factor $\tau = 1.2$ such that the rate of exchange in the experiment is roughly $30\% \sim 40\%$. For the dynamic parameters, the additive Gaussian intensity $c = 0.1$ and the step size $\epsilon = 5 \times 10^{-6}$ in (20). To propose a new sample, the dynamics will simulate a trajectory of length $N = 200$.

**Model architectures.** The RNN for Fashion-MNIST contains one LSTM layer [19] as the first layer, with the input/output dimensions of $28/128$. It takes as the input via scanning a $28 \times 28$ image

Table 1: Result of Bayesian learning experiments on real datasets.

| | RNN on Fashion-MNIST | | | ResNet on CIFAR-10 | | |
|---|---|---|---|---|---|---|
| % permuted labels | 0% | 20% | 30% | 0% | 20% | 30% |
| Adam | $88.56 \pm 0.13\%$ | $87.93 \pm 0.18\%$ | $87.22 \pm 0.23\%$ | $86.03 \pm 0.12\%$ | $80.08 \pm 0.14\%$ | $77.01 \pm 0.16\%$ |
| momentum SGD | $88.83 \pm 0.11\%$ | $88.05 \pm 0.19\%$ | $87.58 \pm 0.20\%$ | $86.11 \pm 0.12\%$ | $79.35 \pm 0.12\%$ | $77.51 \pm 0.15\%$ |
| SGLD | $89.01 \pm 0.13\%$ | $88.25 \pm 0.14\%$ | $87.85 \pm 0.13\%$ | $87.38 \pm 0.14\%$ | $81.16 \pm 0.13\%$ | $78.19 \pm 0.15\%$ |
| RELD | $89.05 \pm 0.13\%$ | $88.31 \pm 0.14\%$ | $87.92 \pm 0.17\%$ | $87.51 \pm 0.13\%$ | $81.19 \pm 0.12\%$ | $78.26 \pm 0.13\%$ |
| SGHMC | $89.12 \pm 0.12\%$ | $88.23 \pm 0.16\%$ | $87.89 \pm 0.19\%$ | $87.50 \pm 0.14\%$ | $81.37 \pm 0.15\%$ | $78.21 \pm 0.14\%$ |
| SGNHT | $89.33 \pm 0.18\%$ | $88.76 \pm 0.22\%$ | $88.04 \pm 0.19\%$ | $87.96 \pm 0.13\%$ | $82.13 \pm 0.17\%$ | $78.54 \pm 0.18\%$ |
| TACT-HMC | $89.74 \pm 0.13\%$ | $88.83 \pm 0.17\%$ | $88.78 \pm 0.17\%$ | $88.01 \pm 0.13\%$ | $82.28 \pm 0.13\%$ | $79.43 \pm 0.12\%$ |
| RENHD in Alg. 2 | $\mathbf{90.87} \pm 0.12\%$ | $\mathbf{89.45} \pm 0.17\%$ | $\mathbf{89.06} \pm 0.16\%$ | $\mathbf{88.41} \pm 0.12\%$ | $\mathbf{84.48} \pm 0.14\%$ | $\mathbf{82.65} \pm 0.13\%$ |

vertically each line of a time. After 28 scanning steps, the LSTM outputs a representative vector of size 128 into ReLU activation, which is followed by a dense layer of size 64 with ReLU activation. The prediction on 10 categories is generated by softmax activation in the output layer. The ResNet for CIFAR-10 consists of 20 standard residual blocks [18]: each contains two "$2d$-Conv + BatchNorm (BN)" pairs (see BN in [22]), seperated by ReLu. It is then wrapped by an identity shortcut, *i.e.* a residual connection, to calculate the residues. All blocks are cascaded. Final output is from a softmax layer. The accuracy is tested with BN layers in test mode.

**Discussion.** RENHD outperforms all non-tempered baselines by a relatively large margin due to the incorporation of tempering. Even in comparison with TACT-HMC, another tempered sampler, RENHD still maintains better performance due to its higher tempering efficiency: RENHD constantly generates correct samples in parallel to fast $\theta$-space exploration in high temperatures, while TACT-HMC has a sequential tempering procedure so that its exploration has to wait until tempering is roughly finished. Moreover, RENHD has much simpler dynamics and fewer hyperparameters, reducing 60% computation for one step of simulation against TACT-HMC. RELD's performance validates the previous discussion that the Langevin dynamics may not be apt for rapid $\theta$-space exploration due to its random-walk-like updates; this disadvantage even limited the effect of a well-tuned tempering scheme possibly because remote modes have never been reached. Hence, we believe that RENHD is of much more practical interest for its virtue of easy implementation, fast tuning, and high tempering efficiency. The result is summarized in Table 1, where the average accuracy of classification is reported with variances calculated from 10 independent runs; for optimizers, *i.e.* momentum SGD and Adam, random initializations are applied to the same network architecture at the beginning of every single run. It took 2.5 hours for the replica ensemble to find a good mode on a single Titan Xp. On the other hand, although RENHD offers better sampling efficiency, it comes with cost of multiplicative space complexity compared with single-replica methods. We analyze, in Appendix F, that number of replicas has to increase in $\sqrt{d}$ of $\theta$'s dimension to retain a proper acceptance probability during replica exchange.

## 5   Conclusion

We propose a new sampler, RENHD, as the first replica-exchange method applicable to mini-batch settings, which can rapidly draw representative samples from complex posterior distributions with *multiple isolated modes* in the presence of *mini-batch noise*. It simulates a ladder of replicas in different temperatures, and alternating between subroutines of evolving the Nosé-Hoover dynamics using the mini-batch gradient and performing configuration exchange based on noise-aware test of acceptance. Experiments are conducted to validate the efficacy and demonstrate the effectiveness; it outperforms all baselines compared by a significant improvement on the accuracy of image classification with different types of neural networks. The results have shown the potential of facilitating deep Bayesian learning on large datasets where multimodal posteriors exist.

## Broader Impact

This paper proposes a practical solution to Bayesian learning. By simulating a collection of replicas at different temperatures, the proposed solution is able to efficiently draw samples from complex posterior distributions. Consequently, the performance of Bayesian learning is improved. Since Bayesian learning is a common tool for many machine learning problems, the social and ethical impacts of this solution are upon specific applications.

## Funding Disclosure

Yaodong Yang was employed by Huawei R&D UK.

## Footnotes

[3]https://github.com/BIDData/BIDMach

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
