[Supplementary Material · renhd-appendix.pdf]

# Appendix of "Replica-Exchange Nosé-Hoover Dynamics for Bayesian Learning on Large Datasets"

## A   Replica Exchange

Figure 1 illustrates the runtime trajectories of five replicas, in which two subroutines are executed in an alternating scheme. In the following subsections, theoretical bases will be established.

Figure 1: A schematic illustration of the replica-exchange protocol. Lines describe 5 trajectories of dynamics of replicas at different temperatures: horizontal segments represent parallel evolution while intersections is replica exchange.

## B   Hermite polynomials and the derivatives

Table 1 lists the first 3 logistic derivatives of odd orders. For higher orders, a recursive routine [4] is developed for fast computation.

Table 1: An example of Hermite polynomials and the derivatives of $g(z) = 1/[1 + e^{-z}]$.

| order | $H_n\left[u = \sigma^2/4\lambda\right]$ | $q_{\mathscr{L}}^{(2n)}(z)$ in terms of $g$ |
|---|---|---|
| $n = 0$ | $1$ | $g - g^2$ |
| $n = 1$ | $2u$ | $g - 7g^2 + 12g^3 - 6g^4$ |
| $n = 2$ | $4u^2 - 2$ | $g - 31g^2 + 180g^3 - 390g^4 + 360g^5 - 120g^6$ |

## C   Empirical results of Gaussianity test on stochastic gradient when training ResNet

Figure 2 illustrates the trajectories of percentage of Gaussian elements within the stochastic gradients during training ResNet-20; each curve represents a different block of ResNet-20. It is clear that the percentage of Gaussian elements within each blocks are higher than 90%, which indicates the Gaussianity assumption for ResNet are appropriate hypothesis.

## D   Effective potentials of replicas at different temperatures

Figure 3 shows the effective potentials of replicas at different temperatures; the corresponding $\pi_j(\theta_j)$ are then illustrated. It becomes clear that when climbing the increasing ladder of temperatures $\{T_j\}$, $\pi_j(\theta_j)$ moves gradually towards a flat histogram.

Figure 2: Gaussianity of each layer in ResNet-20 during training epochs.

Figure 3: (*colored*) The *left* plot shows the effective potentials for 5 replicas at different temperatures. As temperature rises, the energy barrier at $-7$ reduces, which facilitates the passage. The *right* gives the marginal distributions, moving towards flattened histograms during tempering. The blue curves is the real potential (*left*) and thus the true posterior (*right*) at $T = 1$.

## E  Proof of Theorem 1

*Proof.* We prove the existence of the invariant distributions. The uniqueness follows as a consequence of the assumption on ergodicity.

The Nosé-Hoover dynamics in (3) defines a system of stochastic differential equations, which governs the time evolution of state in a probabilistic way from a microscopic perspective. On the other hand, consider the entire ensemble, *i.e.* the collection of all possible states, its evolution can be characterized statistically from a macroscopic point of view through the time evolution of state distribution $\pi_j(\Gamma_j, t)$. The Fokker-Planck equation [6] translates the stochastic dynamics of state into the differential equation

$$
\begin{aligned}
\dot{\pi}_j(\Gamma_j, t) = & -\partial_{\theta_j}^\top \left[ p_j \pi_j \right] - \partial_{p_j}^\top \left[ f(\theta_j) \pi_j \right] + \partial_{p_j}^\top \left[ \xi_j p_j \pi_j \right] \\
& - \partial_{\xi_j} \left[ (p_j^\top p_j - T_j d) \pi_j \right] + \partial_{p_j}^\top \left[ B \partial_{p_j} \pi_j \right],
\end{aligned}
\tag{1}
$$

which can be solved deterministically or even analytically; the invariant distributions can be indicated by $\dot{\pi}_j(\Gamma_j, t) = 0$.

We presume that the invariant distribution of $\xi$ is separable from that of $\theta_j$ and $p_j$ so that $\pi_j(\Gamma_j) = \pi_j(\xi_j)\pi_j(\theta_j, p_j)$. For the marginal distribution $\pi_j(\theta_j, p_j)$, we consider the typical Boltzmann distribution for the Hamiltonian system $(\theta_j, p_j)$ with the potential $U$ and an additive quadratic kinetic energy $p_j^\top p_j / 2$ (we presume all replicas have unit masses) as is defined for our system:

$$
\pi_j(\theta_j, p_j) \propto \exp\left[ - \left[ U(\theta_j) + p_j^\top p_j / 2 \right] \middle/ T_j \right].
\tag{2}
$$

When solving $\dot{\pi}_j(\Gamma_j, t) = 0$, the Boltzmann $\pi_j(\theta_j, p_j)$ in (2) results in the Hamiltonian dynamics [5]; the first and second terms in (1) therefore cancel with each other. The resulting equation *w.r.t.* $\pi_j(\xi_j)$ is simplified as

$$
\frac{1}{\pi_j(\xi_j)} \frac{d\pi_j(\xi_j)}{d\xi_j} = -\frac{1}{T_j} \left[ \xi_j - \frac{B}{T_j} \right],
$$

which gives the unique solution up to a normalizing constant

$$\pi_j(\xi) \propto \exp\left[-\frac{(\xi_j - B/T_j)^2}{2T_j}\right]. \tag{3}$$

Combining two marginal distributions in (2) and (3), the joint distribution of state is obtained as in (4), which is invariant by construction. □

# F  Well-Tempered Ensemble for replica reduction

In this section, we discuss an "optional" device, the *Well-tempered Ensemble* (WTE) [1], for RENHD. WTE is important, albeit not indispensable, for its use of enhancing the memory efficiency of RENHD by reducing the number of replicas for real-world applications, especially for deep neural networks.

In learning very deep neural networks, it might be the case that the parameters grows to hundreds of millions, or even billions. As the efficiency of RENHD relies on the chance of successful exchanges, and the latter is a function of (potential) energy differences: in our case, it resembles the logistic function $g(\Delta E_{jk})$. For a pair of replicas $(j, k)$, a greater overlap of the energy distributions $\pi_j(E)$ and $\pi_k(E)$ will lead to a better chance on the exchange between $\theta_j$ and $\theta_k$. However, observations reveal that the overlap will decrease in the rate of $1/\sqrt{d}$ when the system size $d$ (*i.e.* the dimension for $\theta \in \mathbb{R}^d$) increases [3]. Therefore, to retain a constant acceptance probability, the number of replicas needs to increase in $\sqrt{d}$. For very large systems, the amount of replicas might be prohibitively large.

WTE manages to reduce the number of replicas by enlarging the energy overlap of replicas. It constructs and then maintains for each replica $j$ a time-dependent biasing potential $A_j^\gamma(E, t)$ with $\gamma > 1$ denoting the *tempering factor*, which is a predefined constant defining the increase of energy overlaps by WTE. Figure 4 illustrates the effect of deploying WTE on a demo model with Gaussian energy distributions; the overlap of energy distributions (of adjacent replicas) is substantially enlarged.

The time evolution of the biasing potential $A_j^\gamma$ in WTE is defined by

$$\frac{dA_j^\gamma(E, t)}{dt} = h \exp\left[-\frac{A_j^\gamma(E, t)}{(\gamma - 1)T_j}\right] \cdot \delta\left[E - U(\theta_j(t))\right], \tag{4}$$

where $\theta_j(t)$ indicates the trajectory of $\theta_j$ at time $t$, $h$ is a constant determining the learning rate of $A_j^\gamma$, and $\delta[\cdot]$ denotes the Dirac delta function. As $\gamma$ is a constant, we hereafter omit it for simplicity. Intuitively, the dynamics (4) gradually builts up a $1d$ landscape $A_j$ for replicas $j$, with the coordinates being the energy $E$, at a rate of $h$. The way it determines where to make such incremental changes is by calculating the potential $U$ at the current configuration $\theta_j(t)$.

It has been shown that $A_j(E, t)$ converges asymptotically [2]. With $A_j(E) := A_j(E, t \to \infty)$, the augmented potential can be defined as $V_j(\theta_j) := U(\theta_j) + A_j(U(\theta_j))$ and the tempered energy distribution reads

$$\tilde{\pi}_j^A(E) \propto \int \delta\left[E - U(\theta_j)\right] e^{-V(\theta_j)/T_j} d\theta_j$$

$$= \left(\int \delta\left[E - U(\theta_j)\right] d\theta_j\right) e^{-\left[E + A_j(E)\right]/T_j}. \tag{5}$$

**Theorem 1** ([1]). *The energy distribution* (5) *of the WTE-augmented replica $j$ with converged $A_j$ satisfies*

$$\tilde{\pi}_j^A(E) \propto \left[\pi_j(E)\right]^{1/\gamma}, \tag{6}$$

*indicating that the fluctuation* $\mathbf{var}[E]$ *w.r.t.* $\tilde{\pi}_j^A$ *is effectively amplified by a factor $\gamma$.*

*Proof.* We firstly recall equation (5). We define the integral in the last equity as the temperature-independent density of states, formulated as

$$N_j(E) := \int \delta[E - U(\theta_j)] d\theta_j \tag{7}$$

Figure 4: Effect of deploying WTE on a set of 5 replicas at different temperatures. On the *left* depicts the real histograms of replicas' energy distributions while the *right* shows their tempered counterparts. With WTE enabled, the energy overlap (*shaded*) of adjacent replicas is greatly enlarged, leading to a better chance for successful exchange and thus a higher efficiency.

---

**Algorithm 1** Replica-exchange Nosé-Hoover dynamics with Well-tempered Ensemble

1: **function** NHDYNAMICS($\{\theta_j\}, \{A_j[\cdot]\}, \{T_j\}, \texttt{model}, \mathcal{D}, |\mathcal{S}|_{\text{nhd}}, N, \epsilon, c, \gamma, h, \Delta$)
2: &emsp;&emsp;&emsp;&emsp;&emsp;&emsp;&emsp;&emsp;&emsp;&emsp;&emsp;&emsp; ▷ NHD length $N$; $\epsilon, c$ in (20); $\gamma, h$ in (4); $\Delta$ for quantizing $A_j$
3: &emsp;**for all** $\{j\}$ **do** &emsp;&emsp;&emsp;&emsp;&emsp;&emsp;&emsp;&emsp;&emsp;&emsp;&emsp;&emsp;&emsp;&emsp;&emsp; ▷ all $j$ running in parallel
4: &emsp;&emsp;$v_j \sim \mathcal{N}(0, T_j\epsilon)$ and $s_j \leftarrow c/T_j$ &emsp;&emsp;&emsp;&emsp;&emsp; ▷ resetting auxiliary variables
5: &emsp;&emsp;**for** $n = \text{RANGE}(1, N)$ **do**
6: &emsp;&emsp;&emsp;$\mathcal{S} \leftarrow \text{NEXTBATCH}(\mathcal{D}, |\mathcal{S}|_{\text{nhd}})$ &emsp;&emsp;&emsp;&emsp;&emsp;&emsp;&emsp; ▷ fetching new mini-batch
7: &emsp;&emsp;&emsp;$E_j \leftarrow \texttt{model}.\text{FORWARD}(\theta_j, \mathcal{S})$ &emsp;&emsp;&emsp;&emsp;&emsp;&emsp;&emsp; ▷ $E_j := U(\theta_j)$
8: &emsp;&emsp;&emsp;$\tilde{f}_j \leftarrow \texttt{model}.\text{BACKWARD}(\theta_j, \mathcal{S})$ &emsp;&emsp;&emsp; ▷ evaluating mini-batch gradient
9: &emsp;&emsp;&emsp;$i \leftarrow \text{QUANTIZE}(E_j)$ &emsp;&emsp;&emsp;&emsp;&emsp;&emsp;&emsp; ▷ indexing $A_j[i]$ for quantized $E_j$
10: &emsp;&emsp;&emsp;$dA_j \leftarrow \big[A_j[i+1] - A_j[i]\big]/\Delta$ &emsp;&emsp;&emsp;&emsp; ▷ approximating $dA_j(E)/dE$
11: &emsp;&emsp;&emsp;$dV_j \leftarrow [1 + dA_j]\tilde{f}_j$ &emsp;&emsp;&emsp;&emsp;&emsp;&emsp; ▷ $V_j := U(\theta_j) + A_j(U(\theta_j))$
12: &emsp;&emsp;&emsp;$v_j \leftarrow v_j + dV_j\epsilon - s_j v_j + \mathcal{N}(0, 2c\epsilon)$ &emsp;&emsp; ▷ additional Gaussian noise added
13: &emsp;&emsp;&emsp;$\theta_j \leftarrow \theta_j + v_j$ and $s_j \leftarrow s_j + \big[v_j^\top v_j/d - T_j\epsilon\big]$ &emsp;&emsp; ▷ simulating NHD in (3)
14: &emsp;&emsp;&emsp;$A_j[i] \leftarrow A_j[i] + h\exp\big[-A_j[i]/(\gamma-1)T_j\big]$ &emsp;&emsp;&emsp; ▷ updating $A_j[\cdot]$ *cf.* (11)
15: &emsp;**return** $\{\theta_j\}, \{A_j[\cdot]\}$

16: MAIN:
17: $\{\theta_j\} \leftarrow \text{RANDN}()$ and $\{A_j[\cdot]\} \leftarrow \text{ZEROS}()$ &emsp;&emsp;&emsp;&emsp;&emsp;&emsp; ▷ initialization
18: $\texttt{args} \leftarrow \big(|\mathcal{S}|_{\text{nhd}}, N, \epsilon, c, \gamma, h, \Delta\big)$ &emsp;&emsp;&emsp;&emsp;&emsp;&emsp; ▷ packing arguments
19: **loop**
20: &emsp;$\{\theta_j\}, \{A_j[\cdot]\} \leftarrow \text{NHDYNAMICS}(\{\theta_j\}, \{A_j[\cdot]\}, \{T_j\}, \texttt{model}, \mathcal{D}, \texttt{args})$
21: &emsp;$\{(j, k)\} \leftarrow \text{RAND}()$ &emsp;&emsp;&emsp;&emsp;&emsp;&emsp;&emsp;&emsp; ▷ sampling a set of replicas to swap
22: &emsp;**for all** $\{(j, k)\}$ **do**
23: &emsp;&emsp;$\text{EXCHANGE}(\theta_j, \theta_k, \texttt{model}, \mathcal{D}, |\mathcal{S}|_{\text{re}}, \sigma_*^2, \lambda, \tilde{q}_{\mathscr{C}})$ &emsp;&emsp;&emsp; ▷ recall Algorithm 1
24: &emsp;&emsp;**if** $\theta_j$ and $\theta_k$ exchanged **then** swap $A_j[\cdot]$ and $A_k[\cdot]$
25: &emsp;$\texttt{samples} \leftarrow \big[\texttt{samples}, \theta_0\big]$ &emsp;&emsp;&emsp; ▷ reweighting needed using (9) or (10)

---

68 &emsp; such that the tempered energy distribution is re-written as

$$\tilde{\pi}_j^A(E) \propto N_j(E)\, e^{-\big[E + A_j(E)\big]/T_j}.$$

69 &emsp; As stated by [1], the equilibrium of biasing potential $A_j^\gamma(U) := A_j^\gamma(U, t \to \infty)$ can be formulated as

$$\begin{aligned}
A_j^\gamma(E) &= -\frac{(\gamma-1)}{\gamma} \cdot \big[-T_j \log \pi_j(E)\big] \\
&= -\frac{(\gamma-1)}{\gamma} \cdot T_j \log \big[N_j(E)e^{-E/T_j}\big]^{-1} + \text{const} \\
&= -\frac{(\gamma-1)}{\gamma} \cdot \big[E - T_j \log N(E)\big] + \text{const},
\end{aligned} \tag{8}$$

70 After WTE has converged, the actual potential is essentially the superposition $U(\theta_j) + A_j^\gamma(U(\theta_j))$ of
71 the biasing potential and the original unbiased one. With (7) and (8), the energy distribution reads

$$\pi_j^A(E) \propto \int \delta\big[E - U(\theta_j)\big] e^{-\big[U(\theta_j) + A_j^\gamma(U(\theta_j))\big]/T_j} \, d\theta_j$$

$$= \left[\int \delta\big[E - U(\theta_j)\big] d\theta_j\right] \exp\left[-\frac{E + A_j^\gamma(E)}{T_j}\right]$$

$$= N(E) \exp\left[-\frac{E + (\gamma - 1)T_j \log N(E)}{\gamma T_j}\right]$$

$$= \big[N(E)e^{-E/T_j}\big]^{1/\gamma} = \big[\pi_j(E)\big]^{1/\gamma},$$

72 which would give an approximately same average $\langle E \rangle$ as in the canonical ensemble but with the
73 fluctuation **var**$[E]$ amplified by a factor of $\gamma$.

74 $\qquad\qquad\qquad\qquad\qquad\qquad\qquad\qquad\qquad\qquad\qquad\qquad\qquad\qquad\qquad\qquad\qquad\qquad\qquad$ □

75 An intuitive example can be obtained when the energy distribution is Gaussian, *i.e.* $\pi_j(E) \propto$
76 $e^{-(E-\langle E \rangle)^2/2T_j}$, the well-tempered distribution is also Gaussian with the exactly same average but larger
77 variance $\pi_j^A(E) = \big[\pi_j(E)\big]^{1/\gamma} \propto e^{-(E-\langle E \rangle)^2/2\gamma T_j}$.

78 The marginal distribution of $\theta_j$ for the WTE-augmented replica $j$ is then modified as (*cf.* (**??**))

$$\tilde{\pi}_j^A(\theta_j) \propto e^{-V_j(\theta_j)/T_j}$$

$$= \exp\left[-\frac{U(\theta_j) + A_j(U(\theta_j))}{T_j}\right] \propto \pi_j(\theta_j) \, e^{-A_j(U(\theta_j))/T_j}, \tag{9}$$

79 which deviates from the concerned marginal distribution $\pi_j(\theta_j)$ in (**??**) by a factor $e^{-A_j(U(\theta_j))/T_j}$. A
80 re-weighting procedure needs to be conducted by simply implementing importance sampling with
81 the same factor. In practical scenarios where WTE is deployed, large models, *e.g.* deep neural
82 networks, often involve; it is usually the canonical average of some function $r(\theta_j)$, *i.e.* its Monte
83 Carlo integration *w.r.t.* $\pi_j(\theta_j)$, rather than the posterior distribution $\rho(\theta|\mathcal{D}) \equiv \pi(\theta \,|\, T = 1)$ itself that
84 really matters. For that average, we can readily evaluate it in a simple and unbiased way derived from
85 (9):

$$\langle r(\theta_j) \rangle_{\pi_j} = \frac{\big\langle r(\theta_j) e^{A_j(U(\theta_j))/T_j} \big\rangle_{\tilde{\pi}_j^A}}{\big\langle e^{A_j(U(\theta_j))/T_j} \big\rangle_{\tilde{\pi}_j^A}}, \quad \text{with samples from } \tilde{\pi}_j^A, \tag{10}$$

86 where the biasing potential $A_j(U(\theta_j))$ can be evaluated on the fly during the simulation.

87 Now we devise WTE's update rule for replica $j$ by setting an array to restore the biasing potential $A_j$.
88 Given the granularity $\Delta$, the energy $E$ is quantized; each segment is then associated to one of the cells
89 in that array. $A_j$ is evaluated for all quantized $E$, with the values registered in the corresponding cells.
90 Time is discretized $t \rightarrow n\Delta t$ using the same steps; the differential equation (4) is hence converted into

$$A_j[E; n] \leftarrow A_j[E_j; n-1] + h\,\delta_{E, E_j^{(n)}} \exp\left[-\frac{A_j[E_j^{(n)}; n-1]}{(\gamma - 1)T_j}\right], \tag{11}$$

91 where the learning rate $h$ controls the size of increments, $\delta_{E, E_j^{(n)}}$ defines the Kronecker delta function

92 in the quantized $E$ while $E_j^{(n)} := U(\theta_j(n\Delta t))$ denoting the potential energy evaluated at the $n$-th step.
93 By initializing $A_j[E; 0] \equiv 0$, the biasing potential is adaptively accumulated through the simulation.
94 Algorithm 1 provides a procedural description of RENHD with WTE deployed..