[Reviews · NeurIPS 2020]

Review 1

Summary and Contributions: This is an interesting, if complicated, paper on using a combination of Nose-Hoover dynamics with replica exchange for training Bayesian models. The paper draws on literature of molecular modelling and applies this to more general Bayesian inference tasks. The methodology is not entirely novel as similar schemes appear already in chemical physics, but the use is new.

Strengths: Even if these types of methods are in use in molecular dynamics, they are not explored in machine learning and may be quite useful in this context. The article represents quite substantial numerical work to implement the indicated scheme.

Weaknesses: Designing the temperature ladder is always a substantial challenge in molecular modelling and is, if anything, even harder in machine learning contexts since we have no physical guidepoints to choose the cutoff temperatures. In practice the hyperparameters representing the large and small temperature will need to be tuned for efficiency and are model dependent. This question is not addressed at all in this artlcle. The numerical method used for NH dynamics with additive noise (adaptive Langevin) within the replica exchange framework of Algorithm 2 could be improved for little cost by better choice of splitting, see Leimkuhler and Shang, SIAM J. Sci Comput. 2016 https://epubs.siam.org/doi/pdf/10.1137/15M102318X

Correctness: It seems to be generally correct, although not all statements have been checked.

Clarity: Yes it is reasonably well written.

Relation to Prior Work: There is a genuine effort cite other works but I think some others could and should be mentioned: http://www2.stat.duke.edu/~scs/Papers/RemdJCP.pdf https://pubmed.ncbi.nlm.nih.gov/26203017/ They should also be carefully discussed in the text.

Reproducibility: Yes

Additional Feedback: Concerns have been raised regarding the ML examples considered here. Although a good submission and interesting, I am swayed that further work could benefit the article in demonstrating the practical value of the approach. Still I think this is a very good article.


Review 2

Summary and Contributions: This paper proposes a sampling methods based on multiple particles following Nosé-Hoover dynamics at different temperatures. The goal is to be able to 1) use the different temperatures to explore the whole state space and discover the modes 2) make it amenable to the large scale setting by mitigating the noise induced by the minibatch. They derive (asymptotic) acceptance tests to exchange the particles at different energy levels to ensure exploration while sampling only at T=1 (i.e. the right distribution). They provide synthetic experiments (well-separated mixture of gaussians) as well as sampling from the posterior of neural networks (on Fashion MNIST and CIFAR-10). ======= I have read the authors' response and lower my score to a 6. 150 epochs should be enough to reach good accuracy on CIFAR-10 and methods are not directly comparable if not properly tuned. In addition, the other reviewers raise good concerns, thus my lowering of the score.

Strengths: The paper does a good job at explaining the challenges of various methods and convincing us that their method makes sense. Furthermore, the method seems to be addressing many of the shortcomings of previous work. Section 2.2 is harder to follow, especially the derivation of the logistic test. The exposition seems correct and complete. They also explain clearly how to choose the hyperparameters of their method (e.g. test and bandwidth).

Weaknesses: One of the main weakness in my opinion is the experimental setup. First of all, on the synthetic example, it seems that a few particles of HMC initialized with a large isotropic gaussian should be able to capture all the modes---even though individual particle might not be able to traverse the low density region. Similarly, HMC with tempering should also be able to do the job. On CIFAR, the "optimization" baselines seems very weak in my opinion. SGD with a standard stepsize schedule and ResNet-18 should reach accuracy > at least 93-94% in less than 300 epochs. As such, it is hard to put the performance of the author's method in context. The fact that SGD/Adam degrades much more in the presence of noise is however encouraging. Another weakness is that, if I understood correctly, the test for the replica-exchange process only gives the right distribution asymptotically. While this can be acceptable in practice, I would have liked to see either under which conditions this is close to the true distribution or an empirical evaluation of the discrepancy.

Correctness: The claims and method seem correct albeit the baselines are a little weak.

Clarity: The paper is very well written and easy to follow. The formatting bothered me as it seems like a lot of white space was unnaturally eaten (e.g. margin between section titles, paragraphs, algorithms, footers etc...). Some equations also seem very tiny (e.g. Eq. (10')) as well as the table and algorithm.

Relation to Prior Work: The work is well put into context and it is clearly explained which variation address which challenges.

Reproducibility: Yes

Additional Feedback:


Review 3

Summary and Contributions: The paper considers the problem of sampling from the posterior distribution in Bayesian inference. To be more precise, the paper approaches the question of stochastic sampling that relies only on minibatches of data at each iteration. To achieve rapid mixing between isolated modes, the authors consider parallel tempered chains and introduce replica-exchange steps into the stochastic Nose-Hoover Dynamics. The crux of this approach is the stochastic test for the replica-exchange step. To develop such a test, the authors follow the paper [An efficient minibatch acceptance test for metropolis-hastings], which introduces the concept of correction distribution. The main contribution of the current paper is the derivation of the analytic formula for the correction distribution. ------ UPDATE ------ I've read the rebuttal and other reviews. Unfortunately, I'm pretty confused by the author's response. In what follows, I'm trying to describe my point of view. The authors propose a kind of parallel tempering algorithm, where they interleave stochastic Nose-Hoover dynamics with the replica-exchange step. The stochastic Nose-Hoover dynamics approximately samples from the target distribution without an acceptance test; hence, it can rely only on minibatches of data. To make this also possible for the replica-exchange, the authors employ Seita's paper [arxiv.org/pdf/1610.06848.pdf] that proposes a general-purpose minibatch modification of Barker's test. The only modification to the Seita's test is the deconvolution formula application to obtain the correction distribution, which is also not novel [https://projecteuclid.org/download/pdf_1/euclid.aoms/1177706375]. Thus, I see the proposed algorithm as a straightforward combination of several techniques: Nose-Hoover + parallel tempering + Seita's test. However, just to be clear, I don't want to claim that this is a weakness of the paper. I just think that the proposed technique cannot be seen as a stand-alone theoretical result and requires a thorough empirical study. The empirical study is the main weakness of the paper. There are two main issues: 1. Dealing with minibatch noise. Simply speaking, Seita's idea is to represent logistic random variable X_log from Barker's test as a sum of two other random variables: X_log = X_corr + Normal(0,\sigma^2), where X_corr adds up to the normal minibatch noise to obtain X_log. Note, that the variance of X_log is fixed, and Var(X_log) = Var(X_corr) + Var(noise), since the variables are independent. Therefore, if Var(noise) > Var(X_log) we cannot find suitable correction distribution. To overcome this issue, Seita et al. propose adaptively enlarge minibatches to reduce the noise's variance at the cost of the algorithm's data-efficiency. For instance, it could require thousands of objects on a simple task (classification between only two digits on MNIST) to evaluate the test. This is the central issue in Seita's paper, and I wonder why the authors of the current work completely excluded it from consideration. They also use minibatch enlargement but do not compare their approach to others in terms of data efficiency. Moreover, in their response, they claim that the correction distribution can be obtained with arbitrary precision for any variance of the noise. At this point, I'm pretty confused. Am I missing something? 2. Comparison with competitors. Besides clearly undertrained baselines on CIFAR-10, the paper lacks crucial points of comparison. As I mentioned in my review, the sampling from the posterior distribution of neural networks is tightly related to the ensembling methods [Simple and Scalable Predictive Uncertainty Estimation using Deep Ensembles; Snapshot Ensembles: Train 1, get M for free]. Moreover, there is a paper [Cyclical Stochastic Gradient MCMC for Bayesian Deep Learning] that also proposes a stochastic MCMC algorithm, has a similar motivation and has a more thorough empirical study. To sum up, I think although the algorithm sounds promising, the paper lacks proper empirical study.

Strengths: The paper considers the actual problem of sampling from the posterior distribution in the setting of large amounts of data. This problem can be considered as a fundamental approach to inference and is important in machine learning. - The paper's main strength is the analytical derivation of the correction distribution for the stochastic test proposed in [An efficient minibatch acceptance test for metropolis-hastings]. The proposed formula allows for a more fast and more stable evaluation of the correction distribution compared to the original procedure. - Another merit of the paper is the incorporation of the stochastic test into the replica-exchange method. Indeed, the replica-exchange method with tempering allows for fast exploration of the state-space, and, as a result, for more efficient sampling.

Weaknesses: - The lack of discussion on the choice of the bandwidth lambda. Indeed, the bandwidth seems like a crucial parameter of the proposed method. It controls the tradeoff between the accuracy and the computational cost. Some practical guidelines on the choice of lambda would greatly help the reader implement the algorithm and reproduce the results. - lines 242-244. Here the authors describe achieved acceleration compared to the original method by Seita et al. However, this comparison is entirely unclear. For instance, the acceleration is reported "in sampling", when the difference between approaches is in the correction distribution, which can be evaluated preliminary. Considering that this is the only comparison with the original method in the paper, I cannot find this discussion to be sufficient. - Unfortunately, the empirical comparison for the ResNet architecture on CIFAR-10 is not correct. The significant issue here is that the models are clearly undertrained since it is possible to achieve 92% test set accuracy for ResNet-20. Unfortunately, this single fact makes the primary experiment invalid. - There is no description of the inference procedure. MCMC methods are usually applied in deep learning to collect several samples in the weights space and then ensemble predictions over these models. Thus, the authors should compare all the methods for the same number of samples, and describe how they collect these samples (it is too expensive to collect all of them). Moreover, I would expect to see the comparison with competing ensembling approaches [Simple and Scalable Predictive Uncertainty Estimation using Deep Ensembles; Snapshot Ensembles: Train 1, get M for free; Cyclical Stochastic Gradient MCMC for Bayesian Deep Learning]. - The paper does not discuss the main weakness of the correction distribution approach: if the minibatch noise is too big, one cannot find a suitable correction distribution. This issue is directly related to the data-efficiency of the method. Since the method could require additional batches to reduce the variance of the energy estimate, at this point, I would expect a proper comparison with other methods. minor comments: - Throughout the paper, the authors refer to the spectrum of distributions and their characteristic functions (line 128, line 168). I think the text would benefit from the precise definition of this notion or informal description. - in the formulation of Theorem 2: what does the phrase "formally preserves" means? - line 207: typo "seems to complicated" -> "seems to be complicated"

Correctness: The derivations in Theorem 2 seem correct, although I have not checked all the formulas. Unfortunately, the empirical evaluation of the method is not correct, as I describe in the weaknesses.

Clarity: Overall, the paper is clearly written, though it lacks a description of some details.

Relation to Prior Work: The paper describes a large body of the prior work. However, I would suggest to add several ensembling techniques for comparison [Simple and Scalable Predictive Uncertainty Estimation using Deep Ensembles; Snapshot Ensembles: Train 1, get M for free; Cyclical Stochastic Gradient MCMC for Bayesian Deep Learning].

Reproducibility: No

Additional Feedback: I would advice the authors to check the concurrent work "Non-convex Learning via Replica Exchange Stochastic Gradient MCMC". This work approaches the same problem and propose similar empirical evaluation.


Review 4

Summary and Contributions: This work presents a replica-exchange sampler where replicas with different temperatures are run in parallel and exchanged periodically. The noise introduced by mini-batch convergence is prevented via Nose-Hoover dynamics. --UPDATE-- I have read the author's response and other reviews. I still think combination of several ideals namely, Nose-Hoover, parallel tempering and Seita's test (with deconvolution) makes this submission an interesting work however, I also think some concerns raised by other reviewers are also valid as such I increase my score to 7.

Strengths: Providing an easy-to-implement algorithm that can be used in arbitrary Bayesian inference settings with large data sets is the main novelty of this paper. The theory behind this approach seems sound and reasonably novel and its empirical evaluation shows a significant improvement over the baseline while proposing a general Bayesian framework that handles big data is indeed relevant to the NeurIPS community.

Weaknesses: -- UPDATE -- Following the reviewers discussion, it seems that the other methods are not fine-tuned as such the comparisons may not be fair and valid.

Correctness: Up to my understanding the presented approach is correct and sound.

Clarity: yes

Relation to Prior Work: Some related work to tackle the mentioned two problems of sampling-based Bayesian inference using mini-batches is discussed in the introduction but a more expanded discussion of prior work could add to the value of this submission.

Reproducibility: Yes

Additional Feedback:


Review 5

Summary and Contributions: When a target has multiple well-separated modes, MCMC chains suffer from poor mixing and typically get trapped exploring local modes. At higher temperatures, the MCMC chains can more easily traverse modes. Replica-exchange methods leverage this by running multiple copies of the MCMC chain at different temperatures that progressively swap states to improve mixing. However, these methods all require the evaluation of the likelihood function. In the Bayesian context, when working with large data sets, this becomes computationally infeasible, rendering traditional tempering methods, as well as gradient-based MCMC chains intractable. This paper builds upon Luo et al. (2019) which combined ideas from HMC and simulated tempering to create a continuous-time first-order MCMC algorithm that uses stochastic gradient with mini-batches and used Nose-Hoover dynamics to correct for the noise-induced from the mini-batches versus the full dataset. This paper proposes a replica-exchange version of Nose-Hoover dynamics which improves mixing and sampling efficiency similar to how classical replica exchange methods parallelize simulated tempering. The major contribution of this paper is how to propose a swap between replicas in a setting when only mini-batches are available. Update: After discussions with other reviewers and reading the author(s) feedback, I have decided to change my score to a 6. I think the theoretical contributions are interesting and novel for the machine learning literature. However, I feel that the methodology and experiments still need to be further developed as well as a careful study of its limitations. For it would be nice to know  (1) What is lost by using mini-batches versus full gradient in Nose-Hoover and deconvolution (2) What is the error incurred by approximating reversibility by using the barker versus the metropolis hasting acceptance (3) What are the class of problems this method is suited for and which ones is it not (4) a better benchmarking and comparison with other competing methods. I understand this is beyond the scope of this paper, but I think that is a good first step and demonstrates the potential of this work, however, I think there is still a fair bit of work that needs to be done before this is a practical method for general Bayesian inference in big-data context. 

Strengths: -  This is the first instance of replica-exchange method in the literature using the Barker acceptance probability instead of the Metropolis-Hastings for swapping between replicas     - The main contribution to the literature is Theorem 2. This gives a novel and mathematically interesting approach to proposing swaps between replicas using mini-batches while approximately preserving detailed balance. This is a very interesting addition to the replica-exchange literature.     - The fact that this can be implemented makes it the only replica-exchange method that can account for noise with mini-batches for posterior inference with well-separated modes. - This is a fairly theoretically well-grounded paper. It is clear the author(s) is/are mathematically strong. 

Weaknesses: - iid assumption and the reliance on CLT makes this restrictive for hard Bayesian problems in practice.  This seems like a strong assumption since the constant variance is critical to the building the algorithm. It should be noted the author(s) do identify empirically models built on Gaussian noise, where this assumption is unsurprisingly valid.     - The Barker probability is used to because it is analytic and its regularity properties, not for its Monte Carlo efficiency. It was shown that that the Metropolis-Hastings acceptance is optimal in terms of Peskun ordering (Peskun 1973, Tierney 1998). It is not clear what is lost compared to traditional replica exchange methods compared by using this versus MH or another local proposal satisfying g(t)=tg(1/t) as discussed in Zanella (2017).     - It seems that this algorithm is quite sensitive to hyper-parameters variance $\sigma^*$, $\lambda$, and batch size. Are their general guidelines on how to pick these? It should also be noted that replica exchange methods are known to be very sensitive to hyperparameters (temperature ladder, number of rungs, etc), the design choices for tuning rely heavily on the Metropolis-Hasting acceptance probability, eg. the geometric spacing for the Gaussian. To make this practical for real problems, there needs to be new guidelines for tuning that accounts for the Barker acceptance rule. 

Correctness: - Line 84: Not sure why we can assume the variance of $\tilde{f}$ is independent of the parameters $\theta$. It was not clear where in the reference on line 84 justifies this.     - Each step of the proof of theorem 2 seems correct, although admittedly I did not check every line/equation in detail.          - Was figure 1 generated by the author(s), it seems very similar to figure 8 in Falcioni & Deem (1999).     - Line 248-251: I note sure why adding more Gaussian noise and increasing the variance is done, and how it addresses the potential Ergodicity issues of Nose-Hoover dynamics as states by the author(s)?

Clarity: - I think it is overall well written. It clearly states its purpose and provides a good story. - The math is accessible and does not require anything too obscure to follow.     - My only criticism in terms of style is that the language in the paper is written for someone with intuition from the physics literature, which could be confusing for some readers.

Relation to Prior Work: - This paper does a good job of identifying how it differs from the current literature.     - It makes compelling arguments as to why Nose-hoover dynamics is a worthy competitor to stochastic gradient-based Langevin algorithms.     - The key contribution to this paper in my opinion is the novel approach to replica-exchange in the presence of mini-batch data and is unrelated to the nose-hoover dynamics. The Nose-Hoover dynamics is required as an ingredient for the replica-exchange algorithm, but those details were built in Luo et al. (2018).

Reproducibility: Yes

Additional Feedback: I think the general implementation/experiments can be improved, but in my opinion this paper is not about tuning replica-exchange methods, but rather showing it can be done in the large setting. I think it will take quite a bit of further research before it is practical for general purpose problems.

[Author Response · NeurIPS 2020]

We appreciate the reviewers for their valuable comments on the improvement of this paper. The reviews are insightful and constructive; we believe that all issues mentioned in the comments can be properly addressed in the final version.

**Response to Two Common Concerns:**

**1. Performance on CIFAR10.** It is possible that ResNet-20 on the CIFAR10 in our experiment were undertrained due to the early-stopping we applied. In the initial experiment, we setup the maximum training epochs as 1000; but the procedure would be manually stopped when the performance on the validation set had stopped from increasing for 5 consecutive epochs. For each experimental setting, we conducted 5 independent runs; each time the training of ResNet-20 has been finalized within about 150 epochs, *i.e.* it encountered performance plateau after 150 epochs. As for all hyperparameters for baselines, we have taken the recommended settings and then conducted a simple grid search within a small interval to determine the best fit. As we haven't been over-tuning our method and under-tuning the baselines, we still believe the current result reflects some advantageous tendency of our method. Of course, we will redo the experiment and manage to obtain a more thorough understanding in the final version.

**2. Choice of hyperparameters.** In fact, we haven't put much more effort in tuning hyperparameters in our experiment and the result seemed satisfactory: the **batchsizes** are set as default values in a typical day-to-day neural network training; the **threshold** $\sigma_*$ can be readily estimated from the sample variances within the objective function evaluated on a small number of mini-batches from the target dataset. As for the **bandwidth** $\lambda$, it influences the accuracy by governing the quality of approximation on compensation distribution: with higher bandwidth, the approximation becomes more accurate and its computation will in return be more time-consuming; with better approximation, the detailed balance will be better preserved, which will lead to more accurate samples. We have found that given relatively small batch variance (as is in our experiment), the accuracy and complexity can be balanced quite well by setting $\lambda$ to a **moderate value**; also, we observed that the value of $\lambda$ is not a sensitive factor that needs much tuning for better performance. Indeed, the design of **temperature ladders** is challenging in the context of machine learning due to the absence of physical guidance; nevertheless, some of the approaches developed for physics may still be applied safely to machine learning, *e.g.* the geometric layout as is applied in this paper. In general, the configuration of a ladder will depend on the specific application and also the architecture of the network; it can be optimized through a grid search. According to our observation, ladders of **eight** temperatures allocated by **geometric** factor 0.05 functioned well in all our experiments.

**To Reviewer 1.** *1.* Please see **Common Concerns**. *2.* We've noticed the Leimkuhler splitting in the very beginning. The omission of this scheme in our paper is primarily due to our focus on replica-exchange protocol. Leimkuhler's scheme is interchangeable with the Euler splitting in our algorithm.

**To Reviewer 2.** *1.* Please see **Common Concerns**. *2.* Our RE protocol works under the circumstances where the evaluation of energy function is perturbed by Gaussian noise. No matter whether the RE criterion is Barker's test (as in our proposal) or Metropolis' alternative, the it involves Gaussian deconvolution, which in either case has no exact analytical solution; we have to leverage approximation to make it work. Nevertheless, a more detailed analysis on the discrepancy is beneficial, we will provided it as a complementary section.

**To Reviewer 3.** *1.* Please see **Common Concerns**. *2.* We claim our analytical approximation being more efficient in generating compensation variable $z_C$, where our proposal enables Gibbs sampler whereas Seita's numerical solution needs lookup tables. The latter is way slower than the former. Detailed comparison will be reported as a complementary section. *3.* Please see **Common Concerns**. *4.* We have conducted the comparison in a different manner, we run each method with the same epochs, which we believe reflects the performance in practice. For those baselines with much slower sampling speed, our advantage lies in the time efficiency in real world. These latest ensemble methods will be compared and discussed. *5.* Actually, since Gaussian distribution decays much faster than the logistic, no matter how large the variance Gaussian noise is, in theory, the correction distribution can be obtained at arbitrary precision. In practice, we predefined the noise threshold $\sigma_*^2$ in order to simplify the computation: with a fixed $\sigma_*^2$, no need to recompute the correction distribution, we simply compensate the actual variance $\sigma^2$ up to $\sigma_*^2$ and reuse the recomputed numerics. Multiple mini-batches might be required in the rare case the actual variance $\sigma^2$ exceeds the threshold $\sigma_*^2$.

**To Reviewer 4.** Thank you very much for your kind support and endorsement.

**To Reviewer 5.** *1.* We've examined on several latest architectures with residual connections, namely ResNet, DenseNet, Transformer, and Residual LSTM; empirical findings indicate that the gradient noise, no matter how deep a network will be, resembles Gaussian variables. Hence, albeit found in AlexNet, the heavy-tail phenomenon is not a common situation for all neural architectures; at least for some of the latest models, the conventional assumption of Gaussianity is to some extent still valid. Furthermore, all evidences in our experiment support presuming the constant variance in Gaussian noise. The *i.i.d.* assumption relies on the fact that the dataset is built upon examples collected independently from a certain data distribution. *2.* We've noticed that Metropolis criterion is optimal whereas the Barker's alternative is of 70% efficiency compared with the former. The reason Barker's is leveraged is that his proposal is based on logistic distribution, which resembles Gaussian and is super-smooth. It sacrifices some of the efficiency for much smaller discrepancy and much better analytic characteristics. The traditional RE methods with Metropolis' test either fails to address Gaussian noise or encounters severe problems (*e.g.* delta functions) in deriving correction distributions. Zanella's proposal will be examined carefully. *3.* Please see **Common Concerns**.

[Meta-Review · NeurIPS 2020]

The paper proposes a novel MCMC-type algorithm to perform Bayesian inference on large datasets. The paper is a mixture of replica exchange, Nose-Hoover dynamics and non-standard acceptance criterion to deal with mini-batches. All the reviewers participated actively to the discussion after the rebuttal was made available. Although all the ingredients of the proposed method do exist, their combination is original and potentially useful for the ML literature as pointed out by most reviewers. Theorem 2 is also neat and proposes a nice way to propose swaps between replicas using mini-batches. However, the paper suffers from several serious weaknesses which should be addressed in the final version of the paper. There is first a readibility issue. The authors have used savetrees / space saving to some extreme degree. This is bothersome and this was pointed out by reviewers during the dicussion and reported to the PC. The readibility/presentation has to be significantly improved. As pointed out by some referees, there are important highly relevant references that should be included. The final version should also propose guidelines on the selection of lambda and be more explicit about the limitations of the correction distribution approach in the final version. Reviewer 3 points out a signficant limitation ``if the minibatch noise is too big, one cannot find a suitable correction distribution. This issue is directly related to the data-efficiency of the method. Since the method could require additional batches to reduce the variance of the energy estimate, at this point, I would expect a proper comparison with other methods.'' Finally, the final version has to include better baselines and the authors in this respect should follow the recommendations of Reviewer 3. In this current version the ResNet architecture on CIFAR-10 is off. Ensembling methods should be added for comparison as well as techniques such as cyclical stochastic gradient MCMC.